# MAF-Net: Multi-branch Anchor-Free Detector for Polyp Localization and Classification in Colonoscopy

**Xinzi Sun**                                             XINZI_SUN@STUDENT.UML.EDU
**Dechun Wang**                                     DECHUN_WANG@STUDENT.UML.EDU
**Qilei Chen**                                           QILEI_CHEN@STUDENT.UML.EDU
**Jing Ni**                                                     JING_NI@STUDENT.UML.EDU
**Yu Cao**                                                           YCAO@CS.UML.EDU
**Benyuan Liu**                                                     BLIU@CS.UML.EDU
*University of Massachusetts Lowell, USA*

**Shuijiao Chen**                                             4010383@CSU.EDU.CN
**Xiaowei Liu**                                                 LIUXW@CSU.EDU.CN
*Xiangya Hospital of Central South University, China*

## Abstract

Colorectal polyps are abnormal tissues growing on the intima of the colon or rectum with a high risk of developing into colorectal cancer, the third leading cause of cancer death worldwide. The most common types of colorectal polyps include inflammatory, hyperplastic, and adenomatous polyps. Adenomatous polyps are the most dangerous type of polyp with the potential to become cancerous. Therefore, the prevention of colorectal cancer heavily depends on the identification and removal of adenomatous polyps. In this paper, we propose a novel framework to assist physicians to localize, identify, and remove adenomatous polyps in colonoscopy. The framework consists of an anchor-free polyp detection branch for detecting and localizing polyps and a classification branch for global feature extraction and pathology prediction. Furthermore, we propose a foreground attention module to generate local features from the foreground subnet in the detection branch, which are combined with the global feature in the classification branch to enhance the pathology prediction performance. We collect a dataset that contains 6,059 images with 6,827 object-level annotations. This dataset is the first large-scale polyp pathology dataset with both object segmentation annotations and pathology labels. Experiment results show that our proposed framework outperforms traditional CNN-based classifiers on polyp pathology classification and anchor-based detectors on polyp detection and localization.

## 1. Introduction

Colorectal cancer (CRC) is the fourth most commonly diagnosed cancer worldwide (Bray et al., 2018) and the second most common cause of cancer death in the United States (Siegel et al., 2020). Currently, colonoscopy is the most common and effective approach to screening for and preventing colorectal cancer. During a colonoscopy, abnormal growths, such as colorectal polyps will be identified and removed, and a biopsy may be performed on the removed tissue to determine its pathology. Colorectal polyps are the most critical precursor for CRC. While the large majority of polyps will not become cancerous, certain types of polyps are more likely to turn into cancer. The most common types of colorectal polyps are inflammatory, hyperplastic, and adenomatous. Inflammatory polyps, known as pseudo polyps, develop as a reaction to chronic inflammation in the colon and are generally

benign. Hyperplastic polyps are common, small, and considered an extremely low risk to turn cancerous. Adenomatous polyps are the most dangerous type of polyp with the potential to become cancerous. While a small percentage of adenomatous polyps become cancerous, nearly all malignant polyps begin as adenomatous (Winawer et al., 1993, 2003). Therefore, polyp detection and pathology prediction during colonoscopy are critical for CRC screening and prevention.

Over the past decade, Convolutional Neural Networks (CNNs) have risen as an essential tool for image analysis, and the use of CNNs for Computer-Aided Diagnosis (CAD) applied to polyp detection has met reasonable success. Mo *et al.* (Mo et al., 2018) applied a fine-tuned Faster-RCNN (Ren et al., 2015) with VGG-16 (Simonyan and Zisserman, 2014) as the backbone for polyp detection. Shin *et al.* (Shin et al., 2018) proposed a post-learning scheme to enhance the Faster R-CNN (Ren et al., 2015) detector. The post-learning scheme automatically collects hard negative samples and retrains the network with selected polyp-like false positives, which functions similarly to boosting. Sun *et al.* (Sun et al., 2020) proposed an integrated CAD system comprised of a dilated U-Net detector (Sun et al., 2019) and an AFP-Net detector (Wang et al., 2019). The ensemble model achieves an excellent performance in real-world practical use. Zhang *et.al* (Zhang et al., 2019) adopted an SSD (Liu et al., 2016) detector and optical flow to extract temporal information to detect polyp for colonoscopy video. However, these works can only localize polyps in an image with no pathology prediction, mainly due to the lack of available object detection datasets with polyp pathology annotation. Since polyps appearing within an image frame tend to belong to the same type, some studies used CNN-based models as a classifier to extract features and predict polyp histology on the image level (Byrne et al., 2019; Chen et al., 2018; Lui et al., 2020). While these models achieve good performance on pathology classification, they do not provide the location of polyps. Physicians still need to localize the polyp during colonoscopy then remove them.

In this research, we collaborate with the Endoscopy Center of Xiangya Hospital Central South University in China to collect a dataset that contains 6,059 images from 3,115 pathological reports. Both object-level segmentation annotation and pathology category are provided for each polyp. Additionally, we propose a novel multi-task learning, two-branch framework that integrates an anchor-free detection branch and a CNN-based image classification branch for polyp detection and pathology prediction, respectively. Unlike natural images where pre-defined anchors are introduced to tackle the occlusion issue, in medical images such as colonoscopy images, object density is low and occlusions between objects are rare. Therefore, we believe that an anchor-free design that removes the anchor mechanism and represents objects as keypoint is a more suitable solution. We use the object detectors as Region Proposal Networks (RPNs) (Ren et al., 2015) to generate bounding boxes for foreground objects without category prediction and introduce another image classification branch to predict the pathology of polyps in an image. Because the polyps in a certain image usually belong to the same type, we believe that a whole image classifier fits well by extracting global features for pathology prediction. In our framework, two branches share the backbone network to improve the model's generalization performance and reduce the number of parameters. To take advantage of localization information extracted from the object detection branch, we propose a foreground attention module to extract local features, which will be combined with global features to enhance the pathology prediction.

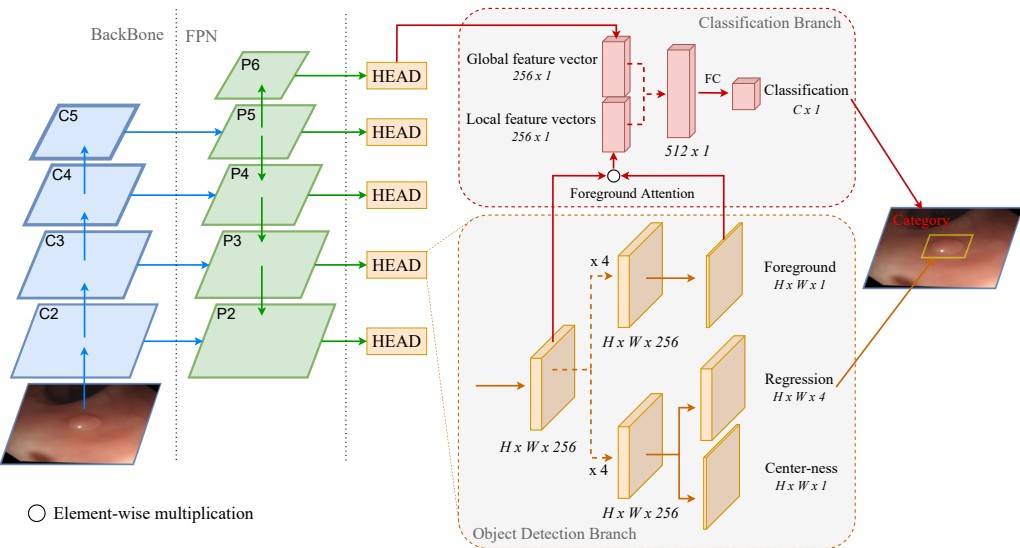

Figure 1: Pipeline of our proposed MAF-Net. The network consists of an object detection branch and an image classification branch.

## 2. Methods

Fig. 1 illustrates the architecture of our proposed framework. It consists of an anchor-free detection branch and an image classification branch for polyp detection and pathology prediction. In this section, we first introduce the anchor-free polyp detection branch. Following that is a brief description of the proposed foreground attention module. Finally, we discuss the details of the classification branch.

### 2.1. Anchor-free Object Detection

The architecture of our anchor-free object detection branch is shown in Fig. 1. Our anchor-free detection heads are single-staged and have similar structures to the heads in FCOS (Tian et al., 2019), where three parallel subnets are dedicated for foreground prediction, object localization, and low-quality detection suppression respectively. A location $(x, y)$ on feature map $F_i \in \mathbb{R}^{H \times W \times C}$ at layer $i$ of the FPN is assigned as a foreground sample if it falls into a ground truth bounding box of any class. Otherwise, the location is considered as a background sample. The foreground subset generates a foreground attention mask where each point $(x, y)$ represents whether this location falls into a ground truth bounding box.

For the regression subset, the network will generate a 4-D tensor for each location on feature map $F_i$. Specifically, for each location $(x, y)$, we have a 4-D regression target tensor $t = (l^*, t^*, r^*, b^*)$ where $l^*, t^*, r^*, b^*$ are the distances from the location to the four sides of the ground truth bounding box it belongs to, formulated as:

$$l^* = x - x_0^l, \ r^* = x_1^l - x, \ t^* = y - y_0^l, \ b = y_1^l - y,$$

where $(x_0^l, y_0^l)$ and $(x_1^l, y_1^l)$ are the top-left and bottom-right corners of the ground truth bounding box. During inference, if a location $(x, y)$ is predicted as a foreground, a prediction

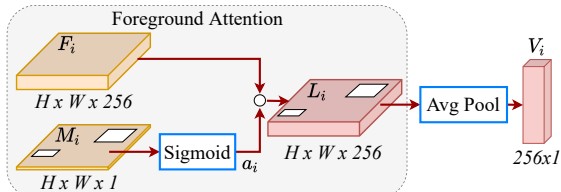

Figure 2: The network architecture of our proposed foreground attention module.

bounding box $(x_0, y_0, w, h)$ will be generated as:

$$x_0 = x - l, \ y_0 = y - t, \ w = l + r, \ h = t + b,$$

where $l, r, t, b$ are the predictions of the regression subnet.

We follow (Tian et al., 2019) to introduce a Center-ness branch in the object detection branch to suppress low-quality bounding boxes produced by locations far away from the center of an object. Different from the original structure, we move the center-ness branch to the regression branch for better localization performance.

### 2.2. Foreground Attention

Recently, attention mechanism has been increasingly applied in semantic segmentation tasks (Fu et al., 2019; Li et al., 2018). Inspired by the success of these works, we propose an attention mechanism to take advantage of the foreground prediction for better polyp pathology classification. In the object detection branch, we generate a foreground prediction for each pixel, which could be viewed as a dense mask of foreground objects. Similar to the position attention module in (Fu et al., 2019), our foreground attention module encodes the contextual information into local feature maps.

Fig. 2 illustrates our proposed foreground attention module. Given a feature map $P_i \in \mathbb{R}^{H \times W \times C}$ at layer $i$ of the FPN, a feature map $F_i \in \mathbb{R}^{H \times W \times C}$ and a foreground mask $M_i \in \mathbb{R}^{H \times W \times 1}$ are generated in the object detection branch. Then, the network applies a sigmoid layer to generate an attention map $a_i \in \mathbb{R}^{H \times W \times 1}$ and performs an element-wise multiplication operation with feature map $F_i$ to generate a local feature map $L_i$. Finally, an average pooling is applied on $L_i$ to obtain a local feature vector $V_i \in \mathbb{R}^{1 \times 1 \times C}$ at scale $i$. Formally,

$$a_i = Sigmoid(M_i), \ L_i = a_i \circ F_i, \ V_i = AvgPool(L_i)$$

Finally, local feature vectors are fed into the classification branch to be combined with global feature vector for pathology prediction.

### 2.3. Classification Branch

Given that polyps appearing in a frame are usually of the same class, most of the studies apply image-level classifiers for pathology prediction (Byrne et al., 2019; Chen et al., 2018; Lui et al., 2020). Follow this idea, we introduce an image-level CNNs-based classification branch for pathology prediction. The classification branch shares the same backbone and FPN with the object detection branch, making the network more parameter efficient than

Table 1: Summary of colonoscopy datasets

| Datasets | Sample | Label | Pathology |
|---|---|---|---|
| CVC-ColonDB (Bernal et al., 2012) | 300 WL images | binary mask | No |
| CVC-PolypHD (Bernal et al., 2012) | 56 HD WL images | binary mask | No |
| CVC-ClinicDB (Bernal et al., 2015) | 612 WL images | binary mask | No |
| ETIS-Larib (Silva et al., 2014) | 196 WL images | binary mask | No |
| Kvasir-SEG (Jha et al., 2020) | 1,000 WL images | binary mask | No |
| SUN (Wang et al., 2018) | 49,799 WL images | bounding box | No |
| IISGS (Nogueira-Rodríguez et al., 2021) | 28,576 WL and NBI images | bounding box | Yes |
| LDPolypVideo (Ma et al., 2021) | 40,266 | binary mask | No |
| Byrne (Byrne et al., 2019) | 106 NBI images | image-level | Yes |
| Chen (Chen et al., 2018) | 284 NBI images | image-level | Yes |
| Lui (Lui et al., 2020) | 100 NBI images | image-level | Yes |
| Hideka (Horiuchi et al., 2019) | 429 WL and NBI images | image-level | Yes |
| Ours | 6059 WL images | Instance segmentation | Yes |

using two independent models to process polyp classification and detection separately. Combining the global feature vector with the local feature vector generated by the foreground attention module, the network provides better pathology classification.

In the pathology classification branch, we first extract a global feature vector by applying average pooling on the deepest feature map $P_6$ of FPN. Meanwhile, five local feature vectors are generated by the foreground attention module simultaneously. Then, local feature vectors are concatenated into a 1280-dimensional vector. After reducing it to a 256-dimensional local feature vector with a 1-d 1x1 convolutional layer, the network combines the local feature vector with the global feature vector by concatenation. Finally, we use a fully connected layer for classification. In this way, the classification branch can take advantage of both global and local information to enhance the pathology classification performance.

## 3. Dataset

While several public datasets are available with polyp segmentation annotation, to the best of our knowledge, there is no public colonoscopy dataset with both localization annotations and pathology information. The existing datasets only contain polyp localization annotations and combine all types of colon polyps into one class. Other in-house polyp pathology datasets merely contain several hundreds of Narrow-Band Imaging (NBI) images with only image-level labels. A summary of colonoscopy datasets is provided in Table 1. We collaborate with the Endoscopy Center of Xiangya Hospital of Central South University in China to collect a large colon polyp pathology dataset. Due to the high cost of NBI devices, their popularity is limited. While model trained on white light (WL) images can be used on any endoscope device at a low cost. Therefore, we only collect WL colonoscopy images.

The resulting colorectal polyp dataset consists of 6,059 colonoscopy images with 6,827 polyp annotations. The colonoscopy images are collected from 3,115 pathology reports obtained from Aug 2014 to Aug 2020. Specifically, the dataset containing 2,456 adenomatous polyps in 2,324 images, 2,810 inflammatory polyps in 2,354 images, and containing 1,561 hyperplastic polyps in 1,381 images. For each polyp, we provide bounding box, polygon segmentation, and pathology annotation. All the polygon segmentation annotations are performed by experienced gastroenterologists from Xiangya Hospital. The resolution of the colonoscopy images varies from $397 \times 352$ to $1,920 \times 1,080$. All the data have been de-identified by data provider and no personal information are exposed.

Table 2: MAF-Net vs ResNet-50 Based Classifier on Image-level Pathology Prediction.

| Method | Adenomatous | | | | Non-adenomatous | | | | Accuracy |
|---|---|---|---|---|---|---|---|---|---|
| | Precision | Recall | F1 | F2 | Precision | Recall | F1 | F2 | |
| ResNet-50 | 71.77 | 62.72 | 66.94 | 64.34 | 77.66 | 84.02 | 80.71 | 82.67 | 75.64 |
| ResNet-50-RGBM | 73.65 | 59.85 | 66.04 | 62.18 | 78.80 | 86.12 | 82.30 | 84.55 | 75.78 |
| MAF-Net-global | 73.95 | 56.98 | 64.36 | 59.72 | 75.73 | 86.99 | 80.97 | 84.48 | 75.19 |
| MAF-Net | 73.27 | 62.91 | 67.70 | 64.74 | 77.98 | 85.13 | 81.40 | 83.60 | 76.39 |

Table 3: Comparison of Confusion Matrix on Image-level.

(a) ResNet-50-RGBM

| True Class | Predicted Class | | |
|---|---|---|---|
| | Adenomatous | Non-Ade | Total |
| Adenomatous | 313 | 210 | 523 |
| Non-Ade | 112 | 695 | 807 |
| Total | 425 | 905 | 1330 |

(b) MAF-Net

| True Class | Predicted Class | | |
|---|---|---|---|
| | Adenomatous | Non-Ade | Total |
| Adenomatous | 329 | 194 | 523 |
| Non-Ade | 120 | 687 | 807 |
| Total | 449 | 881 | 1330 |

Our dataset is the first large-scale polyp pathology dataset with both localization and pathology annotations. The test set reflects the clinical data distribution and helps to obtain a reliable measurement of the system effectiveness. We conduct a statistical analysis on the dataset. We observe that a large fraction (91.7%) of the images only contain one polyp and only 8.3% images contain more than 2 polyps. Note that if an image contains multiple polyps, they are usually of the same category (Wang et al., 2022).

## 4. Experiments and Results

In the early stages of this research, we experiment with a CNN-based image classifier and several object detectors for polyp classification and detection respectively. The results show that object detectors can precisely detect and localize polyps but frequently mislabel them. On the other hand, while the CNN-based image classifier achieves better performance on polyp classification, it does not perform the polyp detection and localization, which reduces its value of practical use. Hence, to take advantage of these two types of models, we propose a novel network that consists of an object detection coupled with a classification branch to provide both polyp localization and classification.

### 4.1. Image Classification

For a fair comparison, we implement the image classifiers based on ResNet-50 (He et al., 2016), which is the same backbone of our proposed model. The first classifier is a vanilla ResNet-50 net. The second classifier consists of two parallel ResNet-50 nets. The first one takes an original image as input to extract the global feature and the second one takes a masked image that only has the polyp area for local feature extraction. Global and local features are then combined for pathology prediction. The experimental results are shown in Table 2. ResNet-50-RGBM denotes our second image classifier taking both original **RGB** and **M**asked images as input. We also present the confusion matrix of ResNet-50-RGBM and our proposed MAF-Net in Table 3.

From Table 2, we observe that our MAF-Net achieves better results than typical image classifiers on image-level polyp pathology classification, outperforming the ResNet-RGBM

Table 4: Comparison Between Well-known Object Detectors.

| Method | Adenomatous | | | | Non-adenomatous | | | | Accuracy |
|---|---|---|---|---|---|---|---|---|---|
| | Precision | Recall | F1 | F2 | Precision | Recall | F1 | F2 | |
| Mask R-CNN | 56.47 | 63.69 | 59.86 | 62.10 | 51.60 | 74.13 | 60.85 | 68.18 | 66.75 |
| SOLO | 65.46 | 59.49 | 62.33 | 60.59 | 56.87 | 66.74 | 61.41 | 64.50 | 73.28 |
| Faster R-CNN | 66.73 | 63.32 | 64.98 | 63.97 | 58.59 | 76.37 | 66.31 | 72.00 | 74.47 |
| SSD | 66.01 | 61.31 | 63.58 | 62.20 | 59.14 | 73.91 | 65.70 | 70.39 | 72.94 |
| FCOS | 70.07 | 56.59 | 62.61 | 58.85 | 62.65 | 81.24 | 70.74 | 76.69 | 74.94 |
| MAF-Net | 68.40 | 60.04 | 63.95 | 61.54 | 65.64 | 74.24 | 69.68 | 72.35 | 76.18 |

by 0.61% on classification accuracy. Meanwhile, MAF-Net has a better recall, F1, and F2-score on adenomatous polyp classification, which makes up a comparatively small proportion in the dataset. From Table 3, we observe that the MAF-Net can identify more adenomatous polyps that are more dangerous and important for CRC screening and prevention than the ResNet-50-RGBM model. These results demonstrate that local features extracted from the polyp area can effectively help the model better classify the polyp pathology, especially for the adenomatous polyp type that is more dangerous.

### 4.2. Object Detection

In this section, we compare the detection performance of our model with several well-known object detectors, including Faster R-CNN (Ren et al., 2015), SSD (Liu et al., 2016), FCOS (Tian et al., 2019), Mask R-CNN (He et al., 2017), and SOLO (Wang et al., 2020). We follow the evaluation metrics presented in the MICCAI 2015 challenge (Bernal et al., 2017) including recall, precision, F1-, and F2-score for adenomatous and non-adenomatous polyps. From Table 4, we can observe that MAF-Net achieves the second-best F1-score on both adenomatous and non-adenomatous polyp detection. For adenomatous polyp detection, the F-1 score of MAF-Net is 1.03% lower than Faster-RCNN. For non-adenomatous polyp detection, the F-1 score of MAF-Net is 1.06% lower than FCOS. When taking the evaluation results of both classes into account, our MAF-Net outperforms all other object detectors. Meanwhile, the MAF-Net outperforms FCOS, the second best method, by 1.24% on accuracy. Here, the accuracy of the two class classification at object level is different from that of image-level classification. Any adenomatous proposal hitting the adenomatous ground truth box will be counted as a true positive. Any adenomatous polyp proposal hitting non-adenomatous ground truth bounding box will be counted as a false positives. Vice versa for the non-adenomatous proposals. A proposal is ignored if it dose not hit any ground truth bounding box.

### 4.3. Discussion

**Anchor-free:** To verify the effectiveness of the anchor-free detector on polyp detection, we experiment with different types of object detectors on polyp detection and compare their performance. In these experiments, we combine three types of polyp into one single class. Table 5 shows the results that demonstrate anchor-free design is a more suitable solution for polyp detection task. Compared with instance segmentation models (SOLO, Mask R-CNN), two-staged object detector (Faster R-CNN), and single-staged object detector (SSD), our anchor-free based polyp detector achieves the best performance on polyp detection.

Table 5: Comparison Between Well-known Object Detectors on One-class Object Detection.

| Method | Precision | Recall | F1-score |
|---|---|---|---|
| Mask R-CNN | 76.81 | 86.68 | 81.45 |
| SOLO | 80.55 | 82.79 | 81.66 |
| Faster R-CNN | 82.66 | 94.31 | 87.88 |
| SSD | 84.03 | 93.13 | 88.35 |
| MAF-Net | 87.77 | 90.15 | 88.94 |

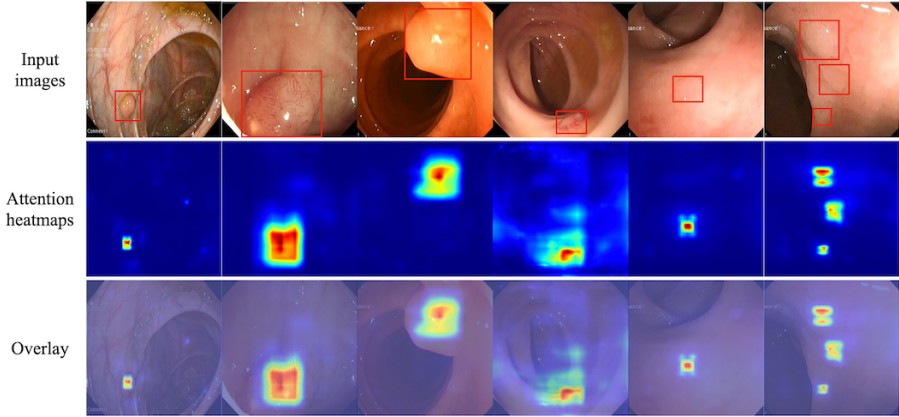

Figure 3: Heatmaps of the foreground attention module. A polyp is highlighted in red, indicating that the network is focusing at the area when generating local features.

**Foreground Attention and local features:** Traditional CNN-based classifiers use the global feature extracted from the whole image for classification. In our task, the image category depends on the objects that are contained in the image. Hence, we expect local features extracted from the object areas can improve the image classification accuracy. From Table 2, we can observe that both our MAF-Net and ResNet-50-RGBM models outperform their counterparts that only utilize global features for classification, verifying that local features can help the classifier achieve better performance. Figure 3 demonstrates the effectiveness of the foreground attention module. The heatmap of the foreground attention module shows that the module precisely localizes the polyp area, and thus can generate salient local feature representations for the whole image classification.

## 5. Conclusion

In this research, we create a large-scale polyp pathology dataset with object-level bounding box, segmentation annotations, and pathology labels. Additionally, we propose a multi-branch convolutional network for polyp detection and classification. It consists of an anchor-free polyp detection branch for detecting and localizing polyps and a classification branch for global feature extraction and pathology prediction. We also propose a foreground attention module to generate local features that are combined with global feature in the classification branch to enhance the pathology prediction performance. Experiments results show that the anchor-free detector outperforms the anchor-based detectors on polyp detection and local features extract by our proposed foreground attention module can effectively improve the polyp pathology classification performance.

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
