# OpenReview forum: "MAF-Net: Multi-branch Anchor-Free Detector for Polyp Localization and Classification in Colonoscopy"
_MIDL.io/2022/Conference — MIDL 2022_

### Official Review · Reviewer_2uya · 2022-01-24

**Confidence:** 5
**Preliminary Rating:** 3
**Recommendation:** Poster

**Summary:**

The authors prosed a deep-learning architecture for simultaneous localisation and classification of colorectal polyps. The proposed architecture based on anchor-free location prediction with a foreground attention module. For the classification, the architecture uses a extracted feature from location prediction branch. The authors evaluate the proposed architecture with their in-house dataset of 6,059 images with 6,827 object-level annotation.

**Strengths:**

Anchor-free approach with attention mechanism for polyp detection. Using attention mechanism, the architecture removes the setting of priors, that is, anchors. The predicted attention mask can be useful for pathological classification, since information around polyps is important for pathological classification.

**Weaknesses:**

Low technical novelty. The architecture bases on the body of ResNet-50 and the head of FCOS. Adding  the attention mechanism and the classification head looks slice update. Predicted attention heatmaps look not so accurate. Unclear descriptions about the assumption, data preparation, training, model selection and evaluations hinder the manuscript.

**Deanonymize Review:**

no

**Detailed Comments:**

The core of the proposed architecture is anchor-free localisation with the proposed attention mechanism. This approach is interesting, since attention mechanism can play important role and remove prior settings for training. However, the advantage of this approach is unclear from experimental results. For details, please refer to my questions. Furthermore, the author integrates polyp classification between adenoma and non-adenoma polyps into localisation architecture. The advantage of this classification is also unclear. For the pathological classification, magnified or ultra-magnified view of colon surface might be necessary for accurate pathological classification reported in the following papers.  Moreover, the evaluation settings including training and model selection are unclear. Therefore, it does not sound technically.

A.	T. Tamaki, et al., Computer-aided colorectal tumor classification in NBI endoscopy using local features, Medical Image Analysis, https://www.sciencedirect.com/science/article/pii/S1361841512001107?via%3Dihub

B.	G. Wimmer, et al., Directional wavelet based features for colonic polyp classification, Medical Image Analysis,  https://www.sciencedirect.com/science/article/pii/S1361841516000165?via%3Dihub

C.	H. Itoh, et al., Robust endocytoscopic image classification based on higher-order symmetric tensor analysis and multi-scale topological statistics, https://link.springer.com/article/10.1007%2Fs11548-020-02255-3



**Final Rating After The Rebuttal:**

4: Weak Accept

**Justification Of The Final Rating:**

In the replies to my questions, the authors showed an intention to give modification, which can remove unclear settings and explain the advantages of the proposed method. Since this modification improves the submitted manuscript, I decided to change my rating as the final rating after the rebuttal.

**Paper Type:**

both

**Questions To Address In The Rebuttal:**

1.   For anchor-free strategy, the author assumed that “occlusions are rare”, which is described in the third paragraph of introduction, but it is not true. In colonoscopy, many folds occlude polyps and leads overlooking of polyps. This is well-known problem in the polyp-detection task. The advantage of anchor-free strategy is still unclear for me. Is there published paper for supporting the authors' description?

2.   The authors describe “the polyps in a certain image usually belong to the same type” in Introduction. Is there published paper reporting this phenomenon?

3.   How to split the data for training, validation, and testing for experimental evaluations? And, how to select the trained model for the evaluations? If training, validation and testing data share the same polyps, the data leakage can happen and evaluation values can be larger than the actual performance.

4.   As the publicly available large datasets, the following three are reported. Why you did not include these datasets in Table 1?

      A. https://www.sciencedirect.com/science/article/pii/S0016510720346551?via%3Dihub

      B. https://link.springer.com/article/10.1007%2Fs00521-021-06496-4

      C. https://github.com/dashishi/LDPolypVideo-Benchmark

5.   In Table 4, when I compared the Precision and Recall between FCOS and the proposed MAF-Net, the results look just a trade of between two evaluation metrics. Therefore, the advantage of the proposed MAF-NET is unclear for me. How do you think about this point?

6.	Why the authors use the operator of Kronecker product to express Hadamard product? It looks weird.


**Special Issue:**

no

---

### Official Review · Reviewer_PmsT · 2022-01-26

**Confidence:** 5
**Preliminary Rating:** 4
**Recommendation:** Poster

**Summary:**

The authors proposed an algorithm for joint detection and classification of colorectal polyps in colonoscopy WL images. The method leveraged a previously proposed method, PCOS, and modified it by including a classification subnet where local and global features are combined to reach a final classification. The method was evaluated in a private dataset of images with location and class labels and compared with SOTA methods.

**Strengths:**

- The proposed method provides a joint polyp characterization and detection. Commonly, these two tasks are addressed separately.
- The addition included to the PCOS approach is sound. However, its benefit would need to be properly demonstrated (see weaknesses)
- The solution is clinically relevant, both detection and polyp type prediction can help reaching an early intervention.
- The solution is evaluated in a large dataset and compared with SOTA methods


**Weaknesses:**

- The method is mostly based on PCOS network but includes a classification subnet that merged global and local features. However, it is not clear substantiated (also during the evaluation) why this is done. The authors indicated "the network provides better
pathology classification while making the network more parameter efficient". The first claim (better pathology classification) is partly addressed by comparing the performance with a network (MAF-Net-global) with only using global information (at least this what I understood, see comment below). However, it would be also good to compare with the original PCOS implementation, where also classification is obtained but using only local information, following the authors' terminology. In this case a classification would be obtained per detected object but this wouldn't be a problem to combine it for a final image classification. The second claim (more parameter efficient) is not demonstrated.
- The evaluation did not show clear benefits of the proposed approach, neither in the detection nor the classification tasks). In Table 2 and 4 there is not a clear tendency to better performance (significant difference is not reported either), with different trends depending on the metric and the task.
- Although the dataset used for evaluation is large and relevant for the task, it would be nice to get a sense of the performance also in publicly available datasets/challenges. This would allow a direct comparison with the current SOTA performance. It is clear that a dataset with both detection and classification tasks is not currently publicly available; but there are for the individual tasks.


**Deanonymize Review:**

no

**Detailed Comments:**

- I would suggest to provide a better explanation of notation. For example, C is used as number of features maps in 2.2. but also as number of classes.
- In Table 2 MAF_Net_Global is introduced but in the text there is not explanation about this approach. Also, in discussion the term MAF-Net-global-local appears for the first time. Please, explain clearly in text and tables the terms used.



**Final Rating After The Rebuttal:**

4: Weak Accept

**Justification Of The Final Rating:**

I don't think that the authors addressed the comments that I provided (A comparison with original FCOS for object detection and classification using solely local features, comparison in publicly available datasets/challenges for specific tasks). However, I still consider the proposed approach is interesting and relevant for MIDL.

**Paper Type:**

methodological development

**Questions To Address In The Rebuttal:**

- A substantiation of the benefits of the proposed approach compared to PCOS. The first claim (better pathology classification) is partly addressed by comparing the performance with a network (MAF-Net-global) with only using global information (at least this what I understood, see comment below). However, it would be also good to compare with the original PCOS implementation, where also classification is obtained but using only local information, following the authors' terminology. In this case a classification would be obtained per detected object but this wouldn't be a problem to combine it for a final image classification. The second claim (more parameter efficient) is not demonstrated.
- An evaluation in publicly available datasets/challenges. it would be nice to get a sense of the performance also in publicly available datasets/challenges. This would allow a direct comparison with the current SOTA performance. It is clear that a dataset with both detection and classification tasks is not currently publicly available; but there are for the individual tasks.

**Special Issue:**

no

---

### Official Review · Reviewer_BYNv · 2022-01-27

**Confidence:** 3
**Preliminary Rating:** 3
**Recommendation:** Poster

**Summary:**

In this work the authors present a multibranch convolutional network for polyp detection and classification. It consists of an anchor free
polyp detection branch for detecting and localizing polyps and a classification branch for global feature extraction and pathology prediction. They also utilize a foreground attention module to generate local features that are combined with global feature in the classification
branch to enhance the pathology prediction performance. Additionally, the authors present a large-scale polyp pathology dataset with object-level bounding box, segmentation annotations, and pathology labels.

**Strengths:**

The biggest strength of the paper is clearly the development of a large-scale polyp pathology dataset with object-level bounding
box, segmentation annotations, and pathology labels.

Then additionally it is an interesting idea to partially split the tasks of object detection and classification in order to benefit the polyp pathology classification performance.

The evaluation against other baselines seems extensive and well performed.

**Weaknesses:**

The actual improvements of the proposed approach over other classification techniques is very small, outperforming by 0.61% on classification accuracy. And for the object detection the imporvement is similarly marginal. So methodoclogically speaking there is not a huge gain by the introduced method over baselines.

It seems that there is no statement regarding patient consent or ethics approval for the dataset collection. This is quite unusual when working wth human data and should be provided. Currently, it only states that "All the data have been de-identified by data provider and no personal information are exposed." which is of course relevant wrt sharing the data further. But they also should be collecte din the first place with ethics approval, I assume?

**Deanonymize Review:**

yes

**Final Rating After The Rebuttal:**

3: Borderline

**Justification Of The Final Rating:**

Thank the authors for the answers to me as well as the other reviewers. You did actually not answer my questions regarding patient consent, which you should keep in mind for future work, especially in more clinically oriented conferences/journals where this is required to provide.
Also, it seems to me as if ithe methodological contributions of this work and its technical novelty is still small.

**Paper Type:**

methodological development

**Questions To Address In The Rebuttal:**

 Why is this dataset alone not being published,e. g. as a data descriptor in something liek Scientifi data so the dataset itselfs becomes cuítable? And will you make this unique dataset available?
It seems like the methodological contributions are rather minor, which should be addressed.
But as stated above the dataset is worth publishing by itself. So I would like you to also adddress if MIDL is the right venue for this work and if you should not rather push the dataset as a separate work in a data journal.

**Special Issue:**

no

---

### Meta-Review · Area_Chair_3799 · 2022-02-21

**Recommendation:** Accept (Poster)
**Confidence:** 5

**Metareview:**

The methodological contributions and technical novelties are relatively small, but the dataset is very useful. Therefore, if the authors want to claim that the dataset is a major contribution, the dataset should be publicly released; otherwise, this contribution cannot be realized.

---

### Decision · Program_Chairs · 2022-02-28

Accept